# Random Walk based Conditional Generative Model for Temporal Networks with Attributes

**Stratis Limnios**
The Alan Turing Institute
London, UK
slimnios@turing.ac.uk

**Andrew Elliott**
Department of Mathematics and Statistics
University of Glasgow &
The Alan Turing Institute, London, UK
Andrew.Elliott@glasgow.ac.uk

**Mihai Cucuringu**
Department of Statistics & Mathematical Institute
University of Oxford &
The Alan Turing Institute, London, UK
mihai.cucuringu@stats.ox.ac.uk

**Gesine Reinert**
Department of Statistics
University of Oxford &
The Alan Turing Institute, London, UK
reinert@stats.ox.ac.uk

## Abstract

We propose a novel method for graph time series generation with node and edge attributes. As graph representations for complex data become increasingly popular, we encounter many time series of graphs with temporal and attribute dependencies, such as communication networks, daily bike rentals or bank transactions. However, the analysis of such graphs can be impeded by privacy or data protection issues, calling for synthetic network time series which serve as surrogate of the observed time series. There are many methods to construct networks, for example transformer-based graph models such as TAGGEN, but it is harder to create data which are faithful to the observed dependencies, including those of attributes across time. Moreover, tabular data generation methods, such as GANs, fail to emulate complex graph structures. To circumvent these limitations, we introduce CTWALK which combines and leverages the strengths of both methods, by coupling TAGGEN that learns the distribution of temporal random walks over the input data, and a conditional tabular GAN (CTGAN) that captures the time dependence of the features. CTWALK is able to mimic edge weight distributions, node labels, and temporal dependencies of the data.

## 1 Introduction

Graphs are ubiquitous; many data sets have an underlying relational structure or inherit graph properties by construction. Graphs then unlock the potential of further data analysis and understanding of data via the extra connectivity and relationships they can exhibit. As graphs are often high-dimensional, machine learning tools for their analysis have been developed. Many domains of application already greatly benefit from graph machine learning tools such as internet network analysis [1], computational biology [2], social network analysis, chemistry [3], transaction and banking data [4] and many others.

Among many tasks, synthetic data generation has received a surge of interest over the recent years. Indeed, many domains, especially in medical and financial applications, benefit and require synthetic data for privacy concerns. In a broader setting, generating large data sets is useful for training complex machine learning algorithms, when real data is hard or expensive to acquire [5].

NeurIPS 2022 Workshop on Synthetic Data for Empowering ML Research.

Many of these methods are variations of Generative Adversarial Neural Networks (GANs) frameworks [6], such as CTGAN [7] which is adapted to tabular data. Static graph generation methods can be traced back to the works of Erdös and Rényi [8]; they rely on families of random graph models, using for instance random or preferential attachment schemes to generate edges. However, their simplicity often fails to model complex dependencies that can be found in real world data; for instance, they often fail to capture heavy-tailed degree distributions that are very common in financial data [9], or a high clustering coefficient often inherent in social networks data.

More recently, many deep learning frameworks have been developed to tackle the complexity of the real world data sets. A first category are Graph Auto-Encoders (GAEs) and their variational versions (GVAEs) [10]. These models generate the edges of the input graph independently, but suffer from complexity concerns as they compute a similarity measure between every pair of nodes to evaluate the likelihood of an existing edge and thus may miss information beyond pairwise interactions.

The second category is based on the propagation paradigm inspired by the message passing step of Graph Neural Networks (GNNs) [11]. By tackling the graph generation problem sequentially, edge dependencies can be tackled more efficiently than by GAEs. Unfortunately, exploiting general graph structure does not scale well in GNNs. Currently, among the most efficient algorithms, in terms of memory scaling and quality of the generated static graphs are GraphRNNs [12], which generate the adjacency matrix entries sequentially using Recurrent Neural Networks (RNNs) [13], and GRAN [14] which is an attention-based GNN that utilizes the already generated part of the graph topology to effectively model complex dependencies between this part and newly added nodes.

Undirected static graphs do not have edge or node attributes, yet already incur scaling problems. For graphs with temporal dependencies and attributes, these problems substantially increase. But indeed, generating synthetic temporal graph data lies at the core of many current applications, including financial transactions, time dependent relational data or traffic data. Some tabular methods can be used to bypass the attribute problems but often miss on the underlying structures. Some models have been proposed also for temporal graph generation, such as TagGen [15], a deep generative model based on generating time dependent random walks using a bi-level multihead attention network to discriminate the realistic walks from the fake ones. Another model is DYMOND [16], a statistical dynamic-graph generative model that samples graphs with realistic structure and temporal node behavior using motifs. Both perform well, but in the scope of our work, they lack scalability, as most of the graph generation methods do, and ignore the existence of potential node or edge attributes.

Our contributions aim to effectively tackle the temporal graph generation problem with potential node and edge attributes, and can be summarized as follows.

- We propose CTWALK to model temporal, directed or not, graphs with attributes, capitalizing on temporal random walks and conditional GANs.
- We illustrate that CTWALK is a scalable solution for temporal graph generation with many attributes.

The paper is structured as follows. In Section 2, we provide notations, background, and details of the problem. Section 3 introduces the CTWALK model. Experimental results are provided in Section 4. Additional results are included in the Appendix.

## 2 Preliminaries

Notations and definitions presented in this section will be used throughout the rest of the paper. We also provide here the essential blocks needed in order to build our deep end-to-end generative model and present in detail the problem which CTWALK will address.

### 2.1 GANs and CTGAN

GANs were initially introduced by [17] for the task of generating synthetic images. The method deploys two neural networks, dubbed as *discriminator* and *generator*. The generator is trained to generate samples similar to the real distribution of the data, while the discriminator network aims to distinguish between real and fake samples. The discriminator achieves the task by maximizing a probability measure between the real distribution and the fake distribution induced by the generator.

Following their inception, GANs have been used in a variety of contexts, with many lines of work aiming to address some of their limitations, including training instability and convergence issues.

In addition to image, speech and text generation, there also has been a surge of interest in employing GANs for tabular data generation [7], as well as time series generation [18, 19]. One such GAN for tabular data generation is CTGAN, a conditional Tabular GAN presented in [7] where every column is modeled as a realization of a multi-modal Gaussian. CTGAN can deal with continuous as well as discrete variables efficiently, while trying to fit these distributions during training. Then each row becomes the concatenation of the continuous variables $[c_1, \ldots, c_k]$, and the one hot encodings of the discrete variables is given by $[d_1^{(1)}, \ldots, d_{i_1}^{(1)} | \ldots | d_1^{(n)}, \ldots, d_{i_n}^{(n)}]$, where $d^{(l)}$ represents one specific discrete variable. This setting will become essential in our problem setup, as we allow for nodes and edges having both discrete and continuous attributes, with the nodes themselves being discrete.

## 2.2 The TAGGEN Model

TAGGEN is a deep generative framework proposed in [15]. It capitalizes on generating temporal random walks on graphs with time stamped edges in order to extract structural and temporal context information from temporal networks.

In order to formalize the notion of temporal co-occurrence and temporal random walk, we use the notations and definitions of [20]. Assume that we observe a time series of $T$ networks $W^{(1 \sim T)}$. We consider a temporal interaction network $\tilde{G}$ as follows. The node set is $\mathcal{V} = \{v_1, \ldots, v_n\}$ and to each $v \in \mathcal{V}$ is associated a bag of temporal occurrences; $v = \{v^{t_1}, v^{t_2}, \ldots\}$. Furthermore, the set of temporal edges is described as $\tilde{E} = \{e_1^{t_{e_1}}, e_2^{t_{e_2}}, \ldots, e_m^{t_{e_m}}\}$. Here, $e_i^{t_{e_i}} = (u_{e_1}, v_{e_1})^{t_{e_1}}$; for an edge $e_1 = (u_{e_1}, v_{e_1})$ to be present at time $t_{e_1}$, both nodes $u_{e_1}$ and $v_{e_1}$ involved in the edge need to be present at time $t_{e_1}$. Then, node occurrences, at time stamp $t_1$ for instance, may be linked by an edge.

A $d$-length temporal random walk is a sequence of incident temporal edges, traversed one after another if they are adjacent in time (within a user-defined time window) and in the same neighborhood.

The TAGGEN pipeline is then three-fold; as an initialization step it passes the overall temporal interaction network as a fixed snapshot to a Deepwalk model, which is a node embedding deep learning model, in order to obtain initial node embeddings. Then it generates a set of fixed length temporal random walks to feed a bi-level self-attention network. The training of this model goes then as follows: for a given temporal random walk, with probability $p_{action} = \{p_{delete}, p_{add}\}$ one performs an addition or deletion action on the random walk. Then the model aims at approximating the maximum likelihood of a synthetic random walk given an action, **Deletion** or **Addition**:

$$p(\tilde{W}_{\text{action}}^{(i)} | W^{(1 \sim l)}) \propto p_{\text{action}}(\text{action}) f_\theta(\tilde{W}_{\text{action}}^{(i)}), \tag{1}$$

where $f_\theta$ is the likelihood of observing $\tilde{W}_{\text{action}}^{(i)}$ given the training data $W^{(1 \sim l)}$. While this model performs reasonably well on a general temporal graph setting when the graph is undirected and unweighted, it does not have the functionality to include attributes of temporal graphs.

**Problem Definition:** Both CTGAN and TAGGEN are considered among the best models in their respective domains of application. However, CTGAN is able to generate tabular data very accurately, but is likely to fail to capture the underlying graph structure and latent temporal dependencies inherent in the data set as each of the rows are sampled independently. On the other hand, TAGGEN is a very powerful tool for temporal graph generation, in particular for temporal edge generation, but it will fail to generate data attributes, either discrete or continuous. Hence, our goal is to provide a hybrid deep end-to-end generation method that is able to replicate temporal graph data with node and edge attributes (without limits on the number of attributes per edge or node), and that scales well with respect to both run time and memory usage.

## 3 CTWALK

To provide a model that is able to generate high-quality graph data with node and edge attributes, we draw on ideas from CTGAN and TAGGEN in order to take advantage of the strengths of both

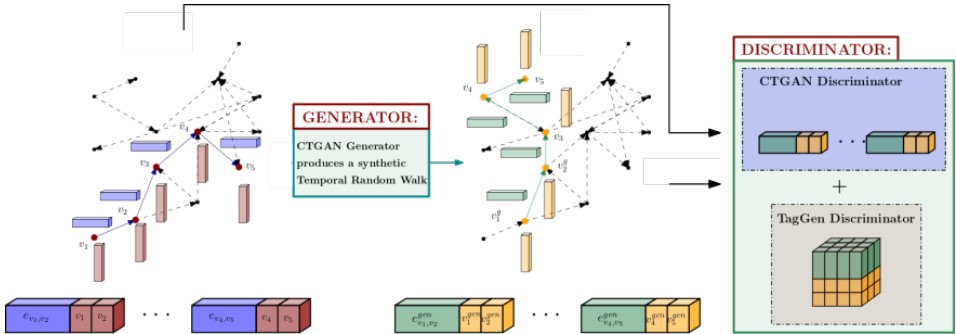

Figure 1: Diagram representing the architecture pipeline on one Temporal Random Walk

methods, while ameliorating their weaknesses. This program has it challenges; CTGAN tries to fit multi-modal Gaussian mixtures representing every feature in the data, whereas TAGGEN has a set of pre-trained embeddings that are concatenated into temporal random walks, which are then used to train a bi-level self attention neural network.

For combining ingredients from both models we first generate and discriminate temporal random walks with attributes using a GAN architecture. Then we communicate attribute level information to a TAGGEN transformer and finally ensure that this is a scalable pipeline while both parts of the model learn together through the same backpropagation.

### 3.1   Model Setup and Initialization

A natural way to share information is to concatenate node and attribute embeddings. Consider the following configuration: assume we have as input a dataset where a row consists of an edge with node and edge attributes. Suppose this edge also exists at a given timestamp $t_1$ between nodes $v_1^{t_1}$ and $v_2^{t_1}$. Now, say we have as a node attribute a vector $w_1$ for node $v_1^{t_1}$ and $w_2$ for $v_2^{t_1}$. Add to that a vector $z_e$ as edge attributes and we finally get the following row as input:

$$x_i = [t_1, w_1, w_2, z_e, v_1^{t_1}, v_2^{t_1}]. \tag{2}$$

This vector is what is traditionally fed to the CTGAN model, where nodes and other discrete variables are converted to one-hot encodings. Conversely, TAGGEN's transformer takes as input a tensor containing the node embeddings occurring in the given input random walk, along with all the embeddings of their other time occurrences for context information.

Hence constructing similar data structures to to feed to both architectures is challenging. The decision is then to start with one representation and adapt it to meet the other representation. We need temporal random walks for graph and temporal context but we also need the attributes.

Suppose we have available a $d$-length temporal random walk $r = \{e_1^{t_1}, e_2^{t_2}, \ldots, e_l^{t_d}\}$; next, we build a feature vector $x_r$ corresponding to this temporal random walk, as follows

$$x_r = [\underbrace{t_1, w_1^1, w_2^1, z_{e_1^{t_1}}, v_1^{t_1}, v_2^{t_1}}_{1^{st} edge}, \underbrace{t_2, w_2^2, w_3^2, z_{e_2^{t_2}}, v_2^{t_2}, v_3^{t_2}}_{2^{nd} edge}, \ldots, \underbrace{t_d, w_{k-1}^k, w_k^k, z_{e_k^{t_d}}, v_{k-1}^{t_d}, v_k^{t_d}}_{d^{th} edge}]. \tag{3}$$

This vector is fed as input to the CTGAN Generator in order to generate a synthetic random walk. Then, following the diagram in Figure 1, we use the CTGAN discriminator to evaluate the newly generated random walk. We also rearrange it as a tensor and add the remaining temporal occurrence embeddings and attribute embeddings of the nodes and edges in the temporal random walk to complete the tensor; these are used for the context information. This tensor is then fed to the TAGGEN, discriminator to evaluate the quality of the temporal random walk structure. Here, instead of one-hot encodings of the nodes in the tensor, we use pre-trained DeepWalk node embeddings. Finally, both discrimination parts yield an overall quality evaluation of the synthetic temporal random walk and its attributes. These evaluations are merged in a joint loss function, defined in the following sections.

**Dimensionality Reduction and Scaling:** As a drawback of our model is the size of the generated tensors, we use a low-rank approximation of this tensor using a parallel factor analysis over the

expected embedding of the edges over the random walks. This results in much smaller tensors, hence a more computational efficient model. Mathematical details for this method can be found in Section 1 of the supplementary material.

**Construction of the Loss Functions:**   Finally, we add the loss functions of the two models both for the generator and the discriminator using the rank approximation for the TAGGEN discriminator;

$$L_{\text{CTWALK}}^{Gen} = L_{\text{CTGAN}}^{Gen}(x) \tag{4}$$

$$L_{\text{CTWALK}}^{Disc} = L_{\text{CTGAN}}^{Disc}(x) + L_{\text{TAGGEN}}^{Disc}(T(x)). \tag{5}$$

Note that we choose not to weight the two losses in the discriminator; in practice we found that weighting did not impact the training positively.

In the following section we present the results of our experiments on the CTWALK model and its advantages as well as its current limitations.

# 4   Experiments

Here we detail the outcome of numerical experiments showcasing the performances of CTWALK on the synthetic data generation task. Testing is conducted on a synthetic data set in order to test CTWALK on triangle motifs over time. In addition, we provide comparisons on two real world use cases across different domains and sizes, namely a citation network and a daily bike rental network.

## 4.1   Datasets

**Triangles:** Our first example is a benchmark dataset which is designed to test the ability of the algorithms to replicate non dyadic structures over time. First we fix a density $d$, number of nodes $n$, and number of time points $t$, and then select $d\,t\,n^2$ sets of 3 nodes uniformly at random. For each set of nodes, we place a temporal directed triangle starting at a time point uniformly selected from $1, \ldots, t-2$ (e.g. for nodes $v_1, v_2, v_3$, we add edges $(v_1, v_2)^{t'}$, $(v_2, v_3)^{t'+1}$, $(v_3, v_1)^{t'+2}$). To add attributes to the graph, we generate a uniform random weight between 0 and 1 to each edge. For the examples in the paper we use 500 nodes to generate this dataset and spread it over 30 days.

**Cycling Dataset:** The Cycling data is a dataset of bike rentals in London, found on the Transport for London platform. It features the bike rentals in London for a week between September 12, 2018 and September 19, 2018. This data set can be represented by a graph featuring 789 stations (nodes) in London linked by a temporal link whenever someone rents a bike and goes from station A to station B. It also has a continuous attribute on the edges, namely the duration of the ride.

**DBLP:** The DBLP dataset consists of papers submitted to conferences over time. We form a network by linking authors if they published a paper together in a given year. As this dataset is initially quite large, for scalability reasons we limit ourselves to the data between 2000 and 2008. We also filter out many conferences with a small amount of papers published. As the nodes are the authors, it features as edge attributes the number of co-authored papers that given year as well as the conference where the given paper forming the edge was submitted. Every node has also as attribute the number of papers the author wrote that year.

**Experimental Setup:** For the experiments we compare the experimental results to the original data as well as to CTGAN. CTGAN, is our benchmark as CTWALK, aims to keep CTGAN's strength in estimating the attributes while adding the ability to learn the complex underlying structure of the graph when fed with the data lists which include adjacency as well as attributes and time stamps. Hence we generate synthetic datasets using our CTWALK model, asking the model to produce $n$ random walks where $n = |E|/k$ with $|E|$ the number of edges in the dataset and $k$ the length of the random walk. In contrast, CTGAN, will produce exactly $|E|$ rows. We report the results in table 1, which includes static graph statistics from the original data set, and fig. 2. In the set of figures 2 we report the statistics of the discrete and continuous attributes generation on the cycling dataset. More results and figures on both datasets can be found in Section 2 of the supplementary material.

| Dataset | #att | Method | #nodes | #edges | #cumul edges | #triangles | assortativity | reciprocity |
|---------|------|--------|--------|--------|--------------|------------|---------------|-------------|
| Triangles | 2 | Orig | 500 | 64.693 | 74.634 | 1.912.563 | −0.0056 | 0.261 |
| | | CTGAN | 500 | 46.857 | 62.427 | 957.312 | −0.0540 | 0.220 |
| | | CTWALK | 500 | 64.099 | 74.593 | 1.898.793 | −0.0081 | 0.267 |
| Cycling | 2 | Orig | 789 | 99.389 | 211.368 | 2.254.888 | 0.082 | 0.547 |
| | | CTGAN | 789 | 99.623 | 150.251 | 3.205.301 | −0.136 | 0.266 |
| | | CTWALK | 789 | 151.632 | 239.563 | 7.458.124 | −0.057 | 0.344 |
| DBLP | 4 | Orig. | 2.121 | 7.645 | 8.049 | 7.220 | 0.163 | 0.163 |
| | | CTGAN | 1.914 | 7.865 | 7.959 | 1.344 | 0.132 | 0.0038 |
| | | CTWALK | 1.742 | 7.940 | 7.989 | 3.455 | 0.056 | 0.0259 |

Table 1: Results for statistics on the benchmark datasets. These statistics are given for the Original dataset (Orig.) the one generated by CTGAN and by CTWALK. Here #att is total number of attributes on the nodes and edges and #cumul edges is the total number of parallel edges.

## 4.2 Discussions and Future Work

Table 1 shows that CTWALK obtains summary statistics of the right order of magnitude. Both CTWALK and CTGAN struggle with identifying the correct direction of assortativity in the Cycling Data set. Our expectation that CTWALK performs at least as well as CTGAN for the attribute generation is confirmed in figure 2 for the DBLP dataset and in the supplementary material for the Cycling dataset. Keeping in mind that CTWALK is designed with the aim to model more efficiently the underlying temporal graph structure, table 1 shows that CTWALK is often closer in graph statistics to the original dataset than CTGAN, thus confirming that CTWALK is an improvement, albeit without having outstanding results. The main advantage of CTWALK is that it handles the temporal graph aspects of the datasets well (results available in Section 2 of the supplementary material). However, even with the low-rank approximation CTWALK struggles to scale above the sizes of these datasets as the one-hot encodings vectors have to be stored in GPU memory to train the CTWALK generator. CTGAN suffers from the same scaling issue as it stems from the scaling of the generator in the GAN architecture, as it aslo produces large one-hot encodings. It is important though to note that where CTWALK fails to scale, CTGAN is very likely to fail as well as it is a generator scaling issue.

**Future Work:** CTWALK is a first step towards tackling our problem and we found that it performs as well as CTGAN to generate good continuous and discrete variables as well as preserving graph and temporal properties. Unfortunately as mentioned above, CTWALK still suffers scalability problems. Future work will include inspiration from the TimeGan model [19]; instead of having a generator producing a giant vector $x_r$, one could pretrain a temporal random walk encoder in parallel with CTWALK while its generator produces the random walk embeddings instead of a whole feature vector $\hat{x}_r$. This pipeline has the potential to drastically reduce the complexity of our model. Although

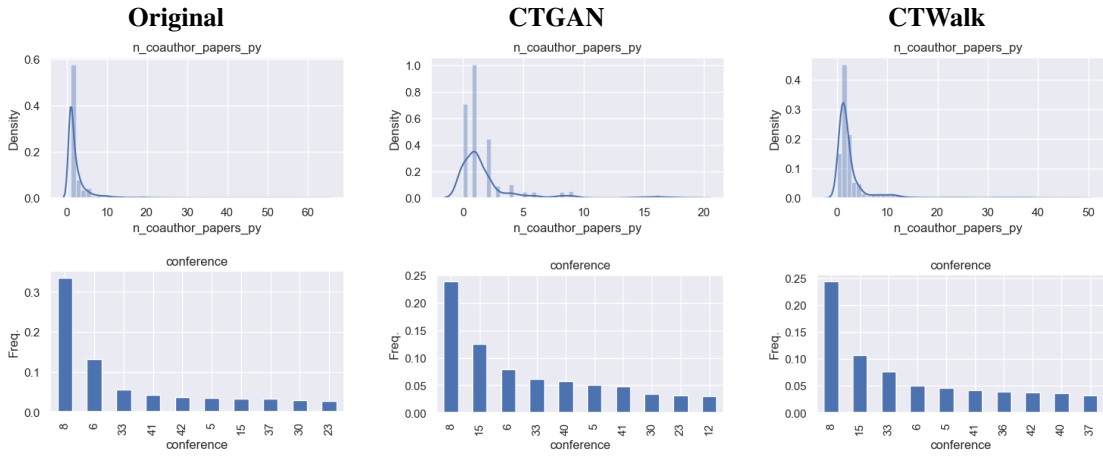

Figure 2: Attributes results comparison for the DBLP data synthetic generation: first row comparing the number of papers per author per year and the second row the number of papers per conference.

this new model would be more intricate and so more complex to train, it may have the advantage of efficiency gain compared to CTWALK.

## 5   Conclusion

In this paper we propose CTWALK, a first approach to the problem of providing a deep end-to-end generation method that is able to replicate temporal graph data with node and edge attributes (without limits on the number of attributes per edge or node), and that scales well with respect to run time and memory usage. As we saw in the experiment section CTWALK is able to produce attributes of state-of-the art level quality without sacrificing graph structure and properties.

However, CTWALK still struggles to handle very large graphs. Improving the scalability by replacing the one-hot encodings by smoother features is a clear next step to be undertaken. Further research will also explore including time series of signed networks.

**Acknowledgements**   This research was supported by The Alan Turing Institute's Finance and Economics Program. We also acknowledge the support of NVIDIA Corporation with the donation of the Titan Xp used for this research. G.R. acknowledges support from EPSRC grants EP/T018445/1, EP/R018472/1, EP/V056883/1, and EP/W037211/1. A.E. is supported in part by EP/W037211/1 and EP/V056883/1 at The Alan Turing Institute.

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
