# OpenReview forum: "Random Walk based Conditional Generative Model for Temporal Networks with Attributes"
_NeurIPS.cc/2022/Workshop/SyntheticData4ML — Neurips 2022 SyntheticData4ML_

### Official Review · Reviewer_tnoo · 2022-10-17
**Combination of conditional tabular GAN and TAGGEN**

**Rating:** 6
**Confidence:** 5

**Review:**

The authors propose to increase the expressive power of TAGGEN, which is a random walk based temporal graph generative model, by combining it with CTGAN, which is a tabular data generation based on GAN. Unlike TAGGEN, the proposed model can generate node and edge attributes as well.

The paper is well structured and easy to read. The only concern with the paper is that, the authors claimed the model is scalable, however, the experiments did not include very large graphs and the number of attributes are relatively small in all experiments.

---

### Official Review · Reviewer_TGMD · 2022-10-18
**This paper introduces a method for generating temporal graphs with node and edge attributes.**

**Rating:** 6
**Confidence:** 3

**Review:**

This paper introduces a method for generating temporal graphs with node and edge attributes.

The model combines CTGAN, a GAN architecture for tabular data, and TagGen, a Transformer-based architecture used to generate temporal random walks on a graph.

The CTGAN generates the temporal random walk, and an additional loss term is added to the discriminator in order to evaluate the quality of the generated walk.

Pros:

* The paper is well-written
* The evaluation is reasonably extensive, with one synthetic dataset and two real-world datasets

Cons:

* Novelty. The method is a fairly straightforward stitching of two existing methods. Some additional work could be made in order to merge them into a single, more sensible architecture for the task.
* The experimental results are not always convincing. From Table 1 it is hard to claim that CTWALK improves over CTGAN. The same is true for Figure 1 in the supplementary. Figure 2 in the supplementary however shows some promising results in modeling some graph statistics.

---

### Official Review · Reviewer_DH4i · 2022-10-19
**Combines CTGAN model with the TagGen model to obtain CTWalk which has both node embeddings as well as temporal graph structure**

**Rating:** 6
**Confidence:** 4

**Review:**

The authors combine two well-known models to solve the synthetic data generation problem for temporal graphs. Results are shown on both synthetic and real-world datasets.

Pros:
(A) They are able to show the new model is better at capturing temporal dependencies over CTGAN
(B) Similar attribute performance as CTGAN i.e. does not sacrifice one for other (other being temporal dependencies).

Cons:

(1) Not much comparison with the TagGen models in terms on temporal dependence.
(2) Does not scale really well and shares the same issues as CTGAN. This is exacerbated in the case of dynamic temporal graphs.

Overall, a good first attempt for combining two models to solve the challenging dynamic graph generation problem.

---

### Meta-Review · Area_Chair_8iNq · 2022-10-19

**Recommendation:** Accept